# Biomarkers for Prostate Cancer Bone Metastasis Detection and Prediction

**DOI:** 10.3390/jpm13050705

**Published:** 2023-04-22

**Authors:** Mingshuai Ying, Jianshui Mao, Lingchao Sheng, Hongwei Wu, Guangchao Bai, Zhuolin Zhong, Zhijun Pan

**Affiliations:** 1Department of Orthopaedic Surgery, The Fourth Affiliated Hospital, International Institutes of Medicine, Zhejiang University School of Medicine, Yiwu 322000, China; 2Department of Orthopaedic Surgery, The Second Affiliated Hospital, Zhejiang University School of Medicine, Hangzhou 310000, China

**Keywords:** predictive biomarkers, prostate cancer, bone metastasis, bone formation/resorption markers, PSA, neuroendocrine markers, liquid biopsy

## Abstract

Prostate cancer (PCa) causes deaths worldwide, ranking second after lung cancer. Bone metastasis (BM) frequently results from advanced PCa, affecting approximately 90% of patients, and it also often results in severe skeletal-related events. Traditional diagnostic methods for bone metastases, such as tissue biopsies and imaging, have substantial drawbacks. This article summarizes the significance of biomarkers in PCa accompanied with BM, including (1) bone formation markers like osteopontin (OPN), pro-collagen type I C-terminal pro-peptide (PICP), osteoprotegerin (OPG), pro-collagen type I N-terminal pro-peptide (PINP), alkaline phosphatase (ALP), and osteocalcin (OC); (2) bone resorption markers, including C-telopeptide of type I collagen (CTx), N-telopeptide of type I collagen (NTx), bone sialoprotein (BSP), tartrate-resistant acid phosphatase (TRACP), deoxypyridinoline (D-PYD), pyridoxine (PYD), and C-terminal pyridinoline cross-linked telopeptide of type I collagen (ICTP); (3) prostate-specific antigen (PSA); (4) neuroendocrine markers, such as chromogranin A (CgA), neuron-specific enolase (NSE), and pro-gastrin releasing peptide (ProGRP); (5) liquid biopsy markers, such as circulating tumor cells (CTCs), microRNA (miRNA), circulating tumor DNA (ctDNA), and cell-free DNA (cfDNA) and exosomes. In summary, some of these markers are already in widespread clinical use, while others still require further laboratory or clinical studies to validate their value for clinical application.

## 1. Introduction

Prostate cancer (PCa), a prevalent cancer in male patients, is also the second-leading cause of death among all malignancies [1,2]. Only 25% of metastatic PCa patients hit the five-year milestone, compared to 99% in localized PCa patients [3,4], which is usually considered to be incurable [1]. Bone metastases (BM) will eventually occur in 80–90% of individuals with advanced PCa [5,6,7]. Advanced PCa is often treated with pharmacological or surgical castration [8,9]. Unfortunately, the majority of individuals who receive castration therapy eventually acquire untreatable castration-resistant prostate cancer (CRPC), which also causes BM [10,11]. Individuals with BM are susceptible to skeletal-related events (SREs), including intolerable bone pain, which greatly lower patients’ quality of life and raises the chance of death [12,13]. The axial skeleton is where PCa metastasizes most frequently, notably in the pelvis or spine [14,15]. The efficient therapy of BM is frequently hampered by the early detection of the condition.

However, it is challenging to detect BM before the emergence of distal clinical symptoms. Traditional techniques, notably MRI, 99mTc-MDP bone scans, CT, or Choline-PET/CT, play critical roles in detecting PCa BM [16]. What’s more, bone tissue biopsy is the most accurate method of detecting BM thus far. However, due to the invasive operation, low specificity, expensive costs, and radiation exposure risk [17], the clinical application of these methods is constrained. Thus, diagnostic and prognostic biomarkers for PCa BM are unavoidably required.

To predict BM in PCa patients, several investigations on possible blood, urine, and tumor tissue-related indicators were conducted. Any of these indicators that might quickly and correctly forecast the onset of BM before being observed by imaging techniques would be helpful to let high-risk individuals get care as soon as possible.

In this paper, we reviewed the current research status of BM markers for PCa, including bone formation markers, bone resorption markers, prostate-specific antigens, neuroendocrine markers, and liquid biopsy markers (Table 1). We provide recommendations on the clinical use of these indicators by critically and objectively assessing their utility in the diagnosis and prognosis of PCa-induced BM.

## 2. Bone Formation Markers

BM is produced as a result of PCa cells interacting with bone cells in both ways. Multiple mechanisms can be used by metastatic PCa cells engrafted in the bone to activate osteoblasts and promote the formation of new bone. When cancer cells emit inflammatory cytokines, including IL-1, IL-6, or PGE2, osteoprogenitor cells and osteoclast precursors are attracted and activated [18]. In the second stage, the osteoclast-resorbed bone matrix is released, unleashing trophic growth factors that draw PCa cells as well as factors involved in matrix remodeling, such as PDGF, IGF-I, and MMPs [19]. Furthermore, PSA- and BMP-release by cancer cells has a direct or indirect effect on bone formation [20]. These elements may encourage the growth of metastatic cancers as well as the production of new bones. By releasing these substances in the metastatic niche, RANKL/OPG ratio adjustments can also be made by PCa cells [21,22]. In general, there is a pathological feedback loop in which healthy bone tissue is gradually replaced by disordered bone deposition, thus more osteoclasts are recruited, and more growth factors are released. This cycle is commonly known as the “vicious cycle” of BM [23].

Unlike breast or lung cancers, PCa BM predominantly features osteoblastic lesions, which involve uncontrollable bone formation [20]. This observation indicates that bone formation markers are crucial in the diagnosis of PCa BM. Studies on bone formation markers are listed in Table 2.

### 2.1. Alkaline Phosphatase (ALP)

The glycoprotein known as ALP is not only present in bones but also in the kidney, gut, placenta as well as the liver. As a low-cost and easily accessible marker, serum ALP has been utilized for decades as a criterion for evaluating PCa BM. It is anticipated it will aid in formulating regular cancer therapy and follow-up plans [38]. The degree of BM is indicated by changes in bone turnover and osteoblastic activity, which are reflected in changes in ALP levels [38,39,40]. Increased levels of ALP were seen in PCa patients with BM, and decreases in ALP levels were also observed in PCa patients with BM after radium-233 radionuclide therapy [41,42]. However, due to its correlation with several non-neoplastic illnesses, including liver cancer and cholecystitis [43], serum total ALP (TALP) shows low sensitivity in the diagnosis of PCa BM, and the cut-off (C/O) value of TALP varies widely in different researches [44]. TALP is better suited for predicting overall survival (OS) than for detecting BM. Clinical trials examining the use of life-extending drugs in metastatic CRPC (mCRPC) have shown that the initial alkaline phosphatase (ALP) level can predict overall survival regardless of the treatment administered [45]. In a meta-analysis, individuals with hormone-sensitive prostate cancer (HSPC) who had high blood ALP levels had an elevated risk of death and disease aggravation [38]. Nevertheless, it has not been demonstrated that TALP can predict survival in PCa patients receiving treatment for hormone-resistant malignancy [46].

Bone-specific alkaline phosphatase (BALP) has shown more potential in various investigations; it is produced by osteoblasts and may also express directly in PCa cells [45] due to its higher specificity [47]. The mineralization of the bone matrix depends on the extracellular enzyme BALP [48]. BALP values are less variable since it is less dependent on food and renal function. Elevated serum BALP is primarily caused by enhanced osteoblastic activity and secondary bone resorption events. These features make BALP a more accurate metric for identifying BM. Zaninotto et al. [24] contrasted 33 healthy individuals with 65 patients who had metastatic PCa, concluding that BALP was more specific (90% vs. 57%) than TALP in diagnosing BM, although both had comparable sensitivity (around 65%).

Patients with BM demonstrated increased TALP and BALP levels compared to those without [25,26,27,49]. In PCa patients, with 18.4 ng/mL as the C/O value, BALP levels exhibited a specificity of 92% and a sensitivity of 92% in BM diagnosis [28]. Furthermore, Karim et al. [29] found from univariate and multivariate analyses that lower BALP levels are a highly important predictor of greater OS in CRPC BM patients. In a meta-analysis of phase III trials of zoledronic acid [50], a higher baseline BAP level in PCa patients predicted more SRE (OR = 1.5; *p* < 0.03) and PCa progression (OR = 2.6; *p* < 0.001). These investigations showed that BALP is a more reliable biomarker in PCa BM diagnosis and prognosis compared to TALP.

### 2.2. Osteocalcin (OC)

OC is the most common non-collagenous protein present in bones. It is synthesized by osteoblasts during osteoid production and is later released into the bloodstream to promote bone growth [51]. Studies have shown that PCa BM patients exhibit higher OC levels [30,47,52], since osteoblasts’ synthesis of OC is very closely tied to the progression of PCa metastasis [52,53]. Thus, elevated serum OC levels are regarded as signs of metastatic disease for PCa. However, Jung et al. [26] reported that OC is rather ineffective as a diagnostic or prognosis indicator of bone metastatic spread. Furthermore, in a comparative study, Maeda et al. [31] found that all markers they mentioned in the text, except for OC, were considerably lower in PCa patients without BM than those with. High lipid amounts that bind OC may also confuse detection [54]. What’s more, previous research has found that circulating OC is highest in early adulthood, lowest in midlife, and has mixed results in older adults [55]. The γ-carboxylated (cOC) and uncarboxylated (ucOC) serve distinct functions in the human body [55,56]. Therefore, OC might not be the best option for predicting BM considering its uncertainty and poor specificity.

### 2.3. Pro-Collagen Type I N-Terminal Pro-Peptide (PINP) and Pro-Collagen Type I C-Terminal Pro-Peptide (PICP)

PINP and PICP are well-established markers of bone formation. Procollagen peptidases contribute to the production of mature type I collagen. This process involves the breakdown of these peptides from opposite ends of the procollagen molecule [57]. Numerous studies have shown that BM can bring up serum PINP and PICP levels in PCa patients [26,32,58,59]. Koopmans et al. [32] demonstrated that a rise in PINP is a sign that PCa BM is progressing. PINP has a specificity of 78% and sensitivity of 68% in BM defemination when the C/O value is set at 58 mcg/L. Significantly, elevated levels of PINP can be observed eight months before the initial detection of metastases through bone scintigraphy [32]. Jung et al. [26] used logistic regression analysis to predict BM, and the overall correct classification for PINP was 84%. According to a single-variable cox regression analysis, PCa BM patients, who present with elevated serum levels of PINP, had shorter survival in contrast to those with normal levels [33]. Piedra et al. [60] considered PINP an adequate predictor of SREs and mortality risk. PINP was similarly thought to be a reliable predictor of OS in PCa BM patients who were receiving zoledronic acid by Jung et al. [61]. The clinical judgment process is therefore aided by the P1NP level, highlighting the crucial roles of PINP and PICP as PCa BM prognostic and predictive markers.

### 2.4. Osteopontin (OPN)

OPN is mainly synthesized by osteoblasts [62]. Some studies have shown that PCa proliferation and metastasis are inextricably correlated to OPN [34,63,64]. Using genetically engineered mice, Khodavirdi et al. revealed that OPN expression is found at various phases of the development of prostate neoplasms, especially in metastatic sites [34]. The enhanced capacity of cancer cells to migrate and metastasize is caused by the upregulation of CD44 and MMP-9 expression, which is triggered by αvβ3 binding OPN [63]. Bisphosphonates could inhibit PCa cells’ migration because OPN binds to the v3 receptor and interacts with CD44 and MMP-9 on the cell membrane [65]. A recent meta-analysis demonstrated that BM and lymph node metastasis were favorably correlated with elevated OPN levels [64]. According to Thoms et al. [35], with regard to the response rate in mCRPC patients following chemotherapy, OPN has the same prognostic relevance as PSA. However, they also found that OPN has limited ability to differentiate metastatic PCa from localized PCa. In breast cancer, OPN-R3, as an OPN RNA aptamer, can minimize local progression and distant metastasis by activating PI3K-Akt-like signaling [66]. In conclusion, OPN still needs more experiments to validate its role in diagnosis, prognosis, and therapeutics of advanced PCa.

### 2.5. Osteoprotegerin (OPG)

OPG can bind to RANKL, impeding its interaction with RANK, thus limiting osteoclast activation and increasing bone mass [67]. The BM of PCa is correlated with the overexpression of OPG. Ye et al. [68] discovered that osteoblasts receive miR-141-3p from PCa cells, which was followed by an increase in OPG expression through p38 MAPK signaling. Osteoblast activity is then stimulated by this progress, which helps create a microenvironment of BM. Al Nakouzi et al. [69] transplanted the IGR-CaP1 cell line into male athymic nude mice that were six weeks old. The researchers discovered that IGR-CaP1-implanted mice exhibited elevated expression of OPG in the bone microenvironment of their tibias. In contrast, control animals that received PBS injections did not exhibit such changes in OPG expression in their tibias. In 2001, Jung et al. [36] had already identified OPG as a biomarker of PCa BM. In this research, the diagnostic sensitivity and specificity of OPG in discriminating PCa patients experiencing BM was 88% and 93%, respectively. Later, Jung et al. [26] compared 10 serum markers in diagnosing PCa BM, and discovered that among all serum markers, OPG, at 3.44 pmol/L, had a specificity of 94% and sensitivity of 93%. Furthermore, in univariate analysis, OPG showed the greatest relative risk, suggesting that it indicates cancer burden rather than being just a predictor of the development of bone density [26]. However, a variety of other organs also produce OPG. Thus, when diagnosing BM, altered serum OPG concentrations brought on by other illness, such as vascular disease [70], must be considered.

Interestingly, several investigations have found that OPG may be involved in the treatment of PCa BM. In a study, PCa cells were injected into severe combined immunodeficient mice both intratibially and subcutaneously, followed by OPG administration [37]. The researchers found that untreated mice had boosted osteoclast counts at the interface of tumor and bone, while OPG-treated mice had regular osteoclast counts. Following OPG treatment, the formation of mixed lytic and sclerotic tibial tumors was thus completely prevented. In another study, OPG inhibited tumor cell growth by decreasing serum PSA levels in a mouse model [71]. In summary, OPG is not only an indicator of bone turnover in PCa patients but also a viable treatment approach for preventing BM of PCa.

## 3. Bone Resorption Markers

### 3.1. Bone Sialoprotein (BSP)

BSP is produced by several cell types, including osteoclasts and hypertrophic chondrocytes [72]. BSP is considered a useful bone resorption diagnostic marker that reflects osteoclast activity [73]. Cancer cells may use BSP to get better migration across matrix barriers as they spread to different tissues, especially bone [74,75]. Bellahcène et al. [76] discovered that among patients with breast cancer, higher BSP expression was linked to BM (*p* = 0.008). Just 8% of patients who tested negative for BSP went on to develop BM, compared to 22% of patients who tested positive for BSP developed BM, according to a retrospective analysis of BSP values in pathological tissue of breast cancer from 454 individuals [77]. High BSP expression may predict BM in PCa, as it did in breast cancer. Waltregny et al. [78] also found high serum BSP levels in individuals with PCa BM. Wei et al. [79] compared BSP’s accuracy with PSA-like indicators in diagnosing PCa BM, demonstrating the higher sensitivity of BSP than other indices. Wang et al. [80] demonstrated that higher BSP levels promote the development of BM. However, Wolfgang et al. [81] found that serum BSP appears to have lower diagnostic potency than all other discovered bone conversion indicators in patients with cancer BM, according to z-score and ROC curve analysis. In addition, according to Jung et al., PCa patients without BM or with lymph node metastases had high serum BSP concentrations as well, much as PCa patients with BM [26]. They hypothesized that excessive BSP expression in PCa tissues would increase serum BSP levels before the occurrence of BM. Jain et al. [82] found that only in the final stage of the illness do serum BSP levels rise in PCa, raising doubts about BSP’s early diagnostic utility in this disease (Table 3). Therefore, unlike in breast cancer, BSP levels should not be regarded as a particular BM marker in PCa diagnosis. Further research should focus on the effectiveness of serum BSP in predicting BM in PCa patients.

### 3.2. C-Telopeptide of Type I Collagen (CTx) and N-Telopeptide of Type I Collagen (NTx)

Approximately 90% of the organic chemical compounds found in bones are made up of type I collagen. As bones are absorbed, CTx and NTx breakdown products of collagen are released and then collected in urine through renal excretion. In CRPC BM patients, lower urine NTx levels often have some connections with more favorable OS in univariate analyses and multivariate analyses [29,83]. Furthermore, solid tumor patients who had elevated or medium NTx levels exhibited a twofold higher risk of SREs and tumor progression [50]. The same outcomes were seen by Jung et al. in BM patients receiving zoledronic acid treatment. As compared to people who had low NTx levels, they found that those who had elevated NTx levels exhibited a 2.21-fold increased chance of bone lesion development, and patients with intermediate NTx levels announced a 1.57-fold greater risk (both *p* < 0.001) [61]. The increases in NTx were observed six months before the occurrence of SREs [61]. Thus, elevated NTx levels may aid in the improvement of the follow-up treatment plan. However, it appears that NTx and CTx have a low sensitivity for diagnosing PCa BM. With a C/O value of 26.9 nmol/L BCE, the sensitivity of NTx is 61%. The sensitivity of CTx is even lower, which drops to 30% with a C/O value of 0.627 μg/L [26]. Intriguingly, Piedra et al. [28] discovered that NTx and CTx both had a 100% sensitivity in the detection of BM in PCa without and with BM or some untreated individuals with benign prostate hyperplasia (BPH). Given the scarcity of relevant studies, the application of CTx or NTx alone in diagnosing BM warrants further investigation.

### 3.3. Tartrate-Resistant Acid Phosphatase (TRACP)

In contrast to CTX and NTX, the level of serum TRACP-5b is unaffected by changes in the time of day, food intake, or renal dysfunction [90]. The specific osteoclast activity marker TRACP-5b is predominantly generated by osteoclasts [91]. TRACP-5b levels have been shown to have significant correlations with BM [26,47]. Jung et al. [26] demonstrated that, at a C/O value of 4.62 U/L, TRACP-5b’s accuracy in diagnosing PCa BM was 77% in sensitivity and 85% in specificity. Yamamichi et al. [84] found that at 335 mIU/dl, TRACP-5b was highly associated with the severity of PCa DM. A combined application of TRACP-5b and PSA can accurately detect PCa BM (AUC = 0.95). According to Ozu et al. [85], patients with BM exhibited significantly greater TRACP-5b levels than those without. Additionally, there was a strong interrelationship between serum TRACP-5b levels and the condition on bone scintigraphy. Salminen et al. [86] demonstrated that at a C/O value of 4.98 U/L, TRACP-5b levels had the highest diagnostic accuracy in distinguishing between individuals with BM and those without. Furthermore, using Kaplan-Meier analysis, they discovered that only 46% of PCa patients with higher TRACP-5b levels (>4.98 U/L) survived for 5 years, whereas for those with lower TRACP-5b levels (<4.98 U/L), 88% of them survived for 5 years (*p* = 0.002). High osteoclast specificity and resistance to hepatic and renal dysfunction are two characteristics of TRACP-5b. Diet and physiological changes also have no impact on it [84]. In conclusion, TRACP-5b is a reliable biomarker for PCa BM diagnosis.

### 3.4. C-Terminal Pyridinoline Cross-Linked Telopeptide of Type I Collagen (ICTP)

ICTP is a byproduct generated during the breakdown of type I collagen [92]. According to the reports of Koga et al. [93] and Koopmans et al. [32], PCa patients with BM had serum ICTP levels that were considerably higher than PCa patients without BM. Wei et al. [79] demonstrated that ICTP showed a sensitivity of 69.05% and specificity of 76.8% for the identification of skeletal metastases at a positive critical value of 4.3 U/L. Another study reveals that ICTP’s sensitivity and specificity in diagnosing BCa BM are 78.6% and 88.0%, respectively, at a C/O value of 5.0 ng/mL [87]. Further, they found that ICTP is more effective at differentiating between PCa patients with and without BM when used in conjunction with PSA and ALP. In research involving 83 samples from 70 PCa individuals (32 with and 38 without BM), the ICTP level showed a stronger correlation with the severity of the condition compared with PSA-like bone markers [31]. Furthermore, Kamiya et al. [88] demonstrated that serum ICTP level was an independent predictor of BM and cause-specific survival. Jung et al. [61] showed that in PCa BM treated with zoledronic acid, ICTP was found to be an appropriate predictor of OS. The ICTP level would be beneficial for both patients, who want complete information about their condition, as well as for clinicians when making decisions about additional treatment protocols. In summary, ICTP is an appropriate biomarker for the treatment of PCa BM, especially when combined with other reliable markers.

### 3.5. Pyridoxine (PYD) and Deoxypyridinoline (D-PYD)

PYD is a cross-link found both within and between collagen molecules that help keep collagen fibers stable and can be deoxidized to form D-PYD cross-linked collagen [94]. With the breakdown of bone tissue, PYD and D-PYD are discharged into the bloodstream [95]. Considerable PYD and DPYD levels were noticed in individuals with PCa BM when compared to those without BM [94,96,97,98]. Furthermore, higher PYD levels were associated with worse OS in a study including 778 PCa patients [89]. Individuals with values above and below the median had different median survival times (15 months vs. 22 months), with a risk ratio of 1.52 (95% CI = 1.28, 1.81). Additionally, because PYD and D-PYD are bone-specific and unaffected by diet, they more accurately depict the degree of BM compared to other bone markers [94].

## 4. Prostate-Specific Antigen (PSA)

PSA is the cornerstone of PCa screening [99]. PSA testing not only performed well in PCa screening, but it also did well in identifying BM in individuals with PCa. Oesterling first hypothesized that PSA provides valuable insights into the prognosis of BM in 1993. They revealed that it does not seem necessary for newly diagnosed PCa patients (without skeletal symptoms) with serum PSA levels equal to or below 10.0 mg/L to have a staging radionuclide bone scan [100]. Salminen et al. [86] constructed ROC curves in 84 PCa patients to compare the efficacy of various markers in detecting BM. They found that PSA had an area under the curve of 0.87, indicating a good degree of diagnostic precision for BM. Kataoka et al. [87] started research involving 155 men with Pca, which reveals that the sensitivity and specificity of PSA in identifying BM are 100% and 79.8%, respectively, at a threshold value of 40.0 ng/mL. Additionally, they discovered that combining PSA, ICTP, and ALP is more effective in identifying individuals with PCa BM from those without. Ozu et al. [85]’s study including 215 untreated PCa patients revealed that PSA, TRACP, and ALP were important independent predictors of BM, with PSA expressing the highest OR through multivariate logistic regression analysis. The predicted likelihood of BM was strongly associated with the actual incidence of BM as calculated by combining PSA, ALP, and TRACP results.

Additionally, a relationship between PSA levels and tumor heterogeneity, tumor size, and disease severity has been documented [101]. However, PSA is only prostate but not PCa specific. Thus, PSA levels can signal innocuous conditions like prostatitis or BPH [99]. In conclusion, PSA is an effective marker for identifying PCa BM. Yet, combining PSA with other markers will improve the accuracy of PCa BM diagnosis.

## 5. Neuroendocrine Markers

An invasive subtype of CRPC [102], the primary diagnostic criteria for NEPC include morphological features and the detection of neuroendocrine markers secreted from neuroendocrine tumor cells, including neuron-specific enolase (NSE) and chromogranin A (CgA) [103,104]. Patients diagnosed with mCRPC displayed 2–3 times higher levels of CgA and NSE compared to those with localized PCa, according to an examination of 1095 serum samples from 395 men, including 157 with localized PCa and 238 with mCRPC [105]. Niedworok et al. [106] proved that CgA levels were considerably higher in advanced PCa patients in comparison to clinically localized cases after examining 110 plasma and 127 serum samples. Before the start of abiraterone acetate, Heck et al. [107] evaluated the NSE and CgA levels at baseline in mCRPC patients. If CgA or NSE levels exceeded the baseline values (85 ng/mL and 16 ng/mL, respectively), OS was considerably reduced. According to Kamiya et al. [108], the mean NSE blood levels were considerably higher in PCa BM patients than in patients without such metastatic conditions (*p* < 0.05), and serum NSE seems to be a standalone death indicator. Furthermore, in some studies, levels of CgA and NSE are also utilized to assess specific effects of chemotherapy and hormonal therapies in CRPC patients [109,110]. However, high CgA levels are also seen in those who have sepsis, cardiac failure, renal failure, hypertension, and several inflammatory diseases [111]. Instead of an absolute value, CgA level may be more informative concerning the earlier stage of PCa.

Another neuroendocrine marker, pro-gastrin releasing peptide (ProGRP), is a reliable indicator for small cell lung cancer [112]. Yu et al. [113] discovered that ProGRP is also useful in diagnosing PCa BM. They observed that the mean ProGRP level in PCa BM patients was 36.81 pg/mL, compared to 22 pg/mL in those without BM. In ROC analysis, the AUC is up to 0.941 when ProGRP was used in combination with total PSA (Table 4).

## 6. Liquid Biopsy Markers

As the gold standard, liquid biopsy for solid tumor diagnosis is less intrusive, more affordable, and provides real-time tumor status information compared to tissue biopsy. Liquid biopsy markers for cancer are classified into three types: cell-free nucleic acids, extracellular vesicles (EVs), and CTCs (Table 5). These methods might help with PCa BM evaluation and prognosis.

### 6.1. MicroRNA (miRNA)

MiRNAs that originate from within the cell interact with mRNA targets to regulate vital cellular processes [129]. MiRNAs have considerable advantages as biomarkers since they are extremely stable and selective for detection in a variety of physiological fluids [130,131]. Numerous pieces of research have shown that miRNAs were involved in the occurrence and progress of PCa BM [132,133,134]. According to Colden et al. [114], upregulation of miR-466 could significantly attenuate the proliferation and BM of PCa cells by regulating the RUNX2 signaling pathway. Compared with normal tissues, miR-466 was significantly downregulated in PCa tissues (*p* < 0.0001), and patients with high miR-466 expression had lower recurrence rates and better prognostic outcomes, making it a great potential for diagnostic and prognostic applications in PCa. Ren et al. [135] discovered that tissues of patients with PCa who display signs of BM showed enhanced miR-210-3p levels than those without BM. Following intracardial injection, miR-210p silencing decreased tumor burden and reduce bone metastatic sites.

In blood circulation, miR-141 level was observed a trend of increasing as the disease develops from organ-confined disease to metastatic PCa [115,121]. Further, the prevalence of BM is inversely correlated with miRNA-141 expression [136]. Thus, miR-141-3p could be monitored to determine risk for PCa-related metastatic disease [137]. Peng et al. [116] discovered that serum miR-218-5p levels were considerably lower in PCa BM than in PCa patients without BM. Their ROC curve analysis in PCa patients, with and without BM, showed an AUC of 0.86 (95% CI: 0.80–0.92, *p* < 0.001). These findings hint that miR-218-5p could be a promising signature molecule for detecting BM in PCa patients. Furthermore, patients with CRPC, which eventually leads to BM, had considerably higher levels of serum miRNA-375, miRNA-378, miRNA-141 [117], and miR-194 [138] than localized PCa. By examining serum samples in PCa BM population, Brase and colleagues [115] also revealed miR-141/375 as the most distinguishing marker for cancer progression.

### 6.2. Cell-Free DNA (cfDNA) and Circulating Tumor DNA (ctDNA)

Cell-Free DNA (cfDNA) are DNA fragments produced mainly by hematopoietic cells undergoing apoptosis [139] entering circulation. In cancer patients, about 1% of total cfDNA is made up of ctDNA [140], which is discharged into circulation when necrosis or apoptosis occurs in malignant cells. Tumor fraction (TFx), or the proportion of ctDNA in plasma cfDNA, together with cfDNA have been suggested as prognostic biomarkers in malignancies, even though it is not a very high percentage [141]. Jung et al. [118] exemplified that cfDNA levels were much higher in patients with metastatic PCa and showed predictive value for OS of metastatic PCa. In order to profile advanced PCa, Morrison et al. [119] paired cfDNA with radiomic analysis of CT bone scans. They discovered a strong correlation between cfDNA and the CT bone scan results. In addition, Kohli et al. [120] found that mCRPC and mHSPC patients with a large amount of ctDNA were observed to have a poor prognosis. For ctDNA, Choudhury et al. [142] proclaimed that TFx level has predictive performance for early response to therapy, which is associated with the extent of visceral and bone metastases in PCa development.

### 6.3. Exosomes

While the aforementioned miRNA and cfDNA demonstrate promising value in PCa BM treatment, the presence of exosomes makes them even more potent biomarkers. Exosomes are EVs with a diameter of 50–150 nm. Various cell types release exosomes that carry cell-derived molecules, such as proteins and nucleic acids [143]. Many miRNA molecules that are biostable in body fluids and blood circulation, and function as cancer biomarkers, have been found in exosomes; some refer to them as exosomal shuttle RNA [144]. However, when RNA is extracted directly from plasma, an excessive volume of platelet-derived RNA may provide a significant background of megakaryocyte RNA, obstructing analysis [145]. Biofluids can be a source for enriching exosomes utilizing tumor-specific surface marker proteins, improving the specificity of the analysis [146]. Furthermore, combining exosomal RNA and ctDNA increases the chance of finding mutated copies in blood samples by up to 10-fold when compared to ctDNA alone [147].

The previously mentioned circulating miR-141 and miR-375 are also part of the exosomal miRNAs that play a diagnostic and prognostic role in PCa BM [125,148]. Ye et al., found both miR-141-3p and miR-375 regulate the BM microenvironment by promoting osteoblast activity, thus inducing BM of PCa [68,149]. Circulating miR-141 levels were significantly higher in metastatic PCa compared to localized PCa [115,121]. Further, the combination of circulating exosomal miR-375 and miR-1290 is an effective predictor of prognosis for CRPC patients [150]. In addition, Laura et al. [122] found that urine exosomal miR-141 and miR-375 levels of patients with high-risk PCa were also significantly higher compared to those of patients with low-risk PCa or normal subjects. Similar to the action of miR-141, exosomal hsa-miR-940 targets and promotes the differentiation of MSCs to osteoblasts and can, by implantation, induce a change from an osteolytic to an osteoblastic phenotype in cancer cells [151]. In contrast to the effect of miR-141, Furesi et al. [152] found that miR-27a-3p, miR-26a-5p, and miR-30e-5p are transferred by PCa-derived EV to suppress osteoblast genesis and prevent bone mineralization, thus possibly aiding in the formation of PCa bone tumors. Exosomal miR-21 has been identified as a promising biomarker for the diagnosis of PCa, but higher plasma exosomal miR-21 has also been observed in metastatic PCa compared to localized PCa [121,153].

Exosomal miR-1246 was described by Bhagirath et al. [123] as having the potential to differentiate between benign, aggressive, and normal types of PCa. They demonstrated that miR-1246 level of aggressive PCa increased 31- and 23-fold in contrast to normal prostatic hyperplasia and BPH, respectively. In addition, high expression of urinary exosomal miR-2909 is also considered an important marker for determining the aggressiveness of PCa [124].

Exosomes of another origin, semen exosomes, are promising non-invasive biomarkers like urine exosomes that can help with the diagnosis and prognosis of PCa BM as well. Semen-derived miR-342-3p and miR-374b-5p are useful in differentiating high-risk PCa [154]. Ruiz et al. [125] also observed that the semen-derived exosomes miR-221-3p, miR-222-3p have high accuracy in determining the prognosis of PCa (AUC = 0.857, *p* = 0.001).

Compared with patients with BPH and non-metastatic PCa, exosomal proteins ITGA3 and ITGB1 were found in higher amounts in the urine of individuals with metastatic PCa according to Bijnsdorp et al. [126]. These proteins may be used as biomarkers in diagnosing PCa metastasis. Other proteins, such as plasma exocrine PSA, have been used to differentiate between BPH and PCa [155]. In another study, PCa-derived exosomes encourage osteoblast proliferation and activity by secreting phospholipase D2, therefore encouraging the development of PCa BM [156]. In conclusion, although some mechanisms need to be further investigated, the various components of the exosome and the combined liquid biopsy are crucial in the diagnosis of PCa BM.

### 6.4. Circulating Tumor Cells (CTCs)

As cancer cells, CTCs could potentially spread to distant sites. Danila et al. [127] found in 120 subjects with progressing CRPC that individuals with BM alone, or with involvement of both bone and soft tissues, had median CTC counts of 10.5 or 13.5 cells, against 2.5 cells in patients without BM, and that baseline CTC was predictive of survival [127]. In addition to CTC count, CTC phenotyping has also been found to be helpful in PCa BM diagnosis and personalized PCa treatment. A strong correlation was observed in the gene expression patterns of nine genes between plasma CTCs and metastatic tissue samples from PCa BM patients in the spine [157]. Another gene, Ezrin, was found to be more expressed in both CTCs and PCa cells with BM characteristics than in those without [128]. Therefore, CTC phenotyping may provide reliable diagnostic and therapeutic targets. However, due to its rarity and tendency to appear in the late stages of cancer, its early diagnostic utility is still debatable.

## 7. Conclusions

Due to delayed diagnosis, patients with PCa BM infrequently obtain successful targeted treatment at an early stage. Thus, it is necessary to investigate more precise diagnostic techniques. The use of BM biomarkers can help predict and detect BM early, thus helping to take effective treatment measures. Numerous studies have discussed the roles of biomarkers in assessing the potency of chemotherapy or immunotherapy in PCa BM patients. Furthermore, since the test sample is typically blood or urine, the markers mentioned in the text are simple to obtain and less invasive to the body. This analysis offers an objective and critical overview of five different classes of promising biomarkers: bone formation/resorption markers, PSA, neuroendocrine, and liquid biopsy markers. Some of these bone formation/resorption markers are well established for the diagnosis of BM from PCa, while others require more clinical studies to determine their ultimate utility in diagnosing PCa BM, although their diagnostic and prognostic value has been described in studies on BM from lung cancer or breast cancer. PSA, a traditional PCa diagnostic marker, also contributes to the diagnosis and prognosis of BM from PCa. Although PSA lacks specificity, serial PSA monitoring is usually beneficial in individuals with advanced PCa. Neuroendocrine and liquid markers have been prominently investigated as emerging biomarkers in recent years, but neither are now utilized frequently in PCa clinical practice. For their clinical validation, more prospective studies are required. PCa BM is a complicated condition. PCa cells evade the primary tumor site, colonize and alter the bone microenvironment, and interact with osteoblasts, osteoclasts, and other bone cells in a vicious loop. Each individual has different biomarkers that are beneficial at different phases of BM’s development. In patients with BM, it is doubtful that a single biomarker would be useful for diagnosis or prognosis. Hence, combining numerous biomarkers and imaging modalities is expected to help the assessment of metastatic PCa.

## Figures and Tables

**Table 1 jpm-13-00705-t001:** Diagnostic and prognostic biomarkers in prostate cancer bone metastases.

Bone formation markers	• Alkaline Phosphatase (ALP)
	• Osteocalcin (OC)
	• Pro-collagen type I N-terminal pro-peptide(PINP)/Pro-collagen type I C-terminal pro-peptide(PICP)
	• Osteopontin (OPN)
	• Osteoprotegerin (OPG)
Bone resorption markers	• Bone sialoprotein (BSP)
	• C-telopeptide of type I collagen (CTx)/N-telopeptide of type I collagen (NTx)
	• Tartrate-resistant acid phosphatase (TRACP)
	• C-terminal pyridinoline cross-linked telopeptide of type I collagen (ICTP)
	• Pyridinoline (PYD) and deoxypyridinoline (D-PYD)
PSA	• Prostate-specific antigen (PSA)
Neuroendocrine markers	• Chromogranin A (CgA)
	• Neurone-specific enolase (NSE)
	• Pro-gastrin-releasing peptide (ProGRP)
Liquid biopsy markers	• MicroRNA
	• Cell free DNA (cfDNA)/Circulating tumor DNA (ctDNA)
	• Exosomes
	• Circulating tumor cells (CTCs)

**Table 2 jpm-13-00705-t002:** Studies on bone formation markers.

Marker	Reference	Sample Location	Sample Size (N)	Finding
ALP	Zaninotto et al. [24]	Serum	65	BALP was more specific (90% vs. 57%) than TALP in diagnosing BM, although both had comparable sensitivity (around 65%).
	Zhao et al. [25]	Serum	792	When BALP level is above 15.55 ng/mL, it has the greatest accuracy for diagnosing PCa BM.
	Jung et al. [26]	Serum	187	At a C/O value of 15.2 ng/mL, BALP’s sensitivity and specificity for diagnosing PCa BM are 75% and 93% respectively.
	Rasch et al. [27]	Serum	111	The sensitivity and specificity of the diagnosis of PCa BM at a mean BALP value of 29.28 ng/mL were 83.8% and 78%, respectively.
	Piedra et al. [28]	Serum	67	In PCa patients, with 18.4 ng/mL as the C/O value, BALP levels exhibited a specificity of 92% and a sensitivity of 92% in BM diagnosis.
	Fizazi et al. [29]	Serum	1901	Lower BALP level (<146 U/L) is a highly important predictor of greater OS in CRPC BM patients.
OC	Arai et al. [30]	Serum	63	OC levels were significantly higher in patients with BM than in those without, and OC responded to endocrine therapy in BM patients.
	Jung et al. [26]	Serum	187	OC is rather ineffective as a diagnostic or prognosis indicator of bone metastatic spread.
	Maeda et al. [31]	Serum	70	Other bone formation/resorption markers, except for OC, were considerably lower in PCa patients without BM than those with.
PINP/PICP	Koopmans et al. [32]	Serum	64	PINP has a specificity of 78% and sensitivity of 68% in BM defemination when the C/O value is set at 58 mcg/L. Elevation of PINP can occur earlier than BM.
	Jung et al. [26]	Serum	187	Logistic regression analysis shows the overall correct classification for PINP to predict BM was 84%.
	Brasso et al. [33]	Serum	153	Those with elevated serum levels of PINP had shorter survival.
OPN	Khodavirdi et al. [34]	Tissue	20	PCa OPN expression showed a gradient increase from early local infiltration to stage of distant metastasis in mice.
	Thoms et al. [35]	Serum	245	OPN is predictive of prognosis in mCRPC patients after chemotherapy. OPN has limited ability to differentiate metastatic PCa from localized PCa.
OPG	Jung et al. [36]	Serum	164	At a C/O value given in the text, the diagnostic sensitivity and specificity of OPG in discriminating PCa patients experiencing BM are 88% and 93%, respectively.
	Jung et al. [26]	Serum	187	OPG, at 3.44 pmol/L, had a specificity of 94% and sensitivity of 93%, respectively.
	Zhang et al. [37]	Serum	30	Prevention of metastatic tibial PCa tumor formation was observed after OPG treatment.

**Table 3 jpm-13-00705-t003:** Studies on bone resorption markers.

Marker	Reference	Sample Location	Sample Size (N)	Finding
BSP	Bellahcène et al. [77]	Tissue	454	Among breast cancer patients, those with high BSP expression exhibit higher metastasis rates.
	Wei et al. [79]	Serum	83	The sensitivity and specificity of BSP for diagnosing PCa BM were 80.95% and 72.8%, higher than other markers mentioned in the text.
	Wang et al. [80]	Serum	356	At a C/O value of 33.26 ng/mL, the sensitivity and specificity for differentiating PCa BM are 78.21% and 79.28%.
	Withold et al. [81]	Serum/Urine	132	Serum BSP have lower diagnostic potency than all other discovered bone conversion indicators in patients with cancer BM.
	Jung et al. [26]	Serum	187	Both lymph node metastases and BM can cause an increase in BSP, thus reducing the diagnostic accuracy.
	Jain et al. [82]	Serum	302	Only in the final stage of the illness do serum BSP levels rise in PCa, raising doubts about BSP’s early diagnostic utility in this disease.
CTx/NTx	Rajpar et al. [83]	Urine	94	In CRPC BM patients, lower urine NTx levels often had some connections with more favorable OS.
	Coleman et al. [50]	Urine	1824	In many cancers, including PCa, high levels of urinary NTx mean a higher incidence of SRE.
	Jung et al. [61]	Serum	52	PCa patients who had elevated NTx levels exhibited higer risk of bone lesion. The increases in NTx were observed 6 months before the occurrence of SREs
	Jung et al. [26]	Serum	187	With a C/O value of 26.9 nmol/L BCE, the sensitivity of NTx is 61%. The sensitivity of CTx is 30% with a C/O value of 0.627 μg/L.
	Piedra et al. [28]	Urine	67	NTx and CTx both had a 100% sensitivity in the detection of BM in PCa without and with BM
TRACP	Jung et al. [26]	Serum	187	At a C/O value of 4.62 U/L, TRACP-5b’s accuracy in diagnosing PCa BM was 77% in sensitivity and 85% in specificity, respectively.
	Yamamichi et al. [84]	Serum	282	Combined application of TRACP 5b and PSA can accurately detect PCa BM (AUC = 0.95).
	Ozu et al. [85]	Serum	215	There is a strong interrelationship between serum TRACP-5b levels and the condition on bone scintigraphy
	Salminen et al. [86]	Serum	84	At a C/O value of 4.98 U/L, TRACP-5b levels had high diagnostic accuracy in diagnosing PCa BM (AUC = 0.82). TRACP-5b predicts OS.
I-CTP	Wei et al. [79]	Serum	83	ICTP showed a sensitivity of 69.05% and specificity of 76.8% for the identification of PCa BM at a positive critical value of 4.3 U/L.
	Kataoka et al. [87]	Serum	155	ICTP’s sensitivity and specificity in diagnosing BCa BM are 78.6% and 88.0%, respectively, at a C/O value of 5.0 ng/mL.
	Kamiya et al. [88]	Serum	222	ICTP has a fairly high accuracy in predicting PCa BM (AUC = 0.85).
	Jung et al. [61]	Serum	52	Cox regression model shows that ICTP was an appropriate predictor of OS.
PYD/D-PYD	Lara et al. [89]	Serum	778	Cox regression analysis reveals that OS was negatively correlated with higher PYD levels.

**Table 4 jpm-13-00705-t004:** Studies on ALP and neuroendocrine markers.

Marker	Reference	Sample Location	Sample Size (N)	Finding
ALP	Oesterling et al. [100]	Serum	2064	It is unnecessary for newly diagnosed PCa patients (without skeletal symptoms) with serum PSA levels equal to or below 10.0 mg/L to have a staging radionuclide bone scan.
	Salminen et al. [86]	Serum	84	PSA has a good degree of diagnostic precision for BM (AUC = 0.87).
	Kataoka et al. [87]	Serum	155	Sensitivity and specificity of PSA in identifying BM are, respectively, 100% and 79.8% at a threshold value of 40.0 ng/mL.
	Ozu et al. [85]	Serum	215	PSA, TRACP, and ALP were important independent predictors of BM. PSA expressing the highest OR through multivariate logistic regression analysis
CgA/NSE	Szarvas et al. [105]	Serum	395	Patients diagnosed with mCRPC displayed 2–3 times higher levels of CgA and NSE compared to those with localized PCa.
	Niedworok et al. [106]	Serum	237	CgA levels were higher in advanced PCa patients than clinically localized cases (45 ng vs. 23 ng/mL, *p* < 0.001;41 vs. 22 ng/mL, *p* = 0.002)
	Heck et al. [107]	Serum	45	OS was considerably reduced when CgA or NSE levels exceeded the baseline values (85 ng/mL and 16 ng/mL, respectively).
	Kamiya et al. [108]	Serum	163	NSE blood levels were considerably higher in PCa BM patients than in patients without PCa BM (*p* < 0.05). Patients with higher NSE levels have poorer survival.
ProGRP	Yu et al. [113]	Serum	163	Mean ProGRP level in PCa BM patients was 36.81 pg/mL, compared to 22 pg/mL in those without BM. ProGRP along with total PSA have high accuracy in diagnosing PCa BM (AUC = 0.941).

**Table 5 jpm-13-00705-t005:** Studies on liquid biopsy markers.

Marker	Reference	Sample Location	Sample Size (N)	Finding
miRNA	Colden et al. [114]	Tissue	96	Compared with normal tissues, miR-466 was significantly downregulated in PCa tissues (*p* < 0.0001), and patients with high miR-466 expression had lower recurrence rates and better prognostic outcomes.
	Brase et al. [115]	Serum/Tissue	674	Both miR-141 and miR-375 levels are elevated in serum or tissue samples of PCa BM patients.
	Peng et al. [116]	Serum/Tissue	223	Serum miR-218-5p levels were considerably lower in PCa BM than in PCa patients without BM (AUC = 0.86).
	Nguyen et al. [117]	Serum	84	Serum MiR-375, miR-378 and miR-141 are significantly elevated in mCRPC patients.
cfDNA/ctDNA	Jung et al. [118]	Serum	184	CfDNA levels were much higher in patients with metastatic PCa and cfDNA has predictive value for OS.
	Morrison et al. [119]	Serum	22	Discovered a strong correlation between cfDNA and the CT bone scan results in metastatic CRPC patients.
	Kohli et al. [120]	Serum	303	mCRPC and mHSPC patients with large percentage of ctDNA were observed a poor prognosis.
Exsomes	Yaman et al. [121]	Serum	51	Levels of miR-141, miR-21, miR-221 were significantly higher in metastatic PCa compared to localized PCa. MiR-141 was the most significant difference (*p* < 0.001; AUC = 75.5%).
	Foj et al. [122]	Urine	162	Urine exosomal miR-141, miR-21 and miR-375 levels of patients with high-risk PCa were significantly higher compared to patients with low-risk PCa or normal subjects.
	Bhagirath et al. [123]	Serum	12	MiR-1246 level differ between benign, aggressive, and normal types of PCa. miR-1246 level of aggressive PCa increased 31- and 23-fold in contrast to normal prostatic hyperplasia and BPH.
	Wani et al. [124]	Urine	210	Levels of miR-2909 can show changes in the aggressiveness of PCa.
	Ruiz et al. [125]	Semen	97	Semen-derived exosomes miR-221-3p, miR-222-3p have high accuracy in determining the prognosis of PCa (AUC = 0.857, *p* = 0.001).
	Bijnsdorp et al. [126]	Urine	13	Exosomal proteins ITGA3 and ITGB1 were found in higher amounts in the urine of individuals with metastatic Pca.
CTC	Danila et al. [127]	Serum	120	Individuals with BM had a CTC counts of 10.5 cells, against 2.5 cells in patients without BM, and baseline CTC was predictive of survival.
	Chen et al. [128]	Serum/Tissue	80	Ezrin was found to be more expressed in both CTCs and PCa cells with BM characteristics than in those without.

## Data Availability

Data sharing is not applicable to this article as no new data were created or analyzed in this study.

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
