# Peer review of "Biomarkers for Prostate Cancer Bone Metastasis Detection and Prediction"

_jpm, 2023, doi:10.3390/jpm13050705_

Round 1
Reviewer 1 Report
The manuscript, “Biomarkers for prostate cancer bone metastasis detection and prediction" is a compilation of many studies linking to various molecular markers with prostate cancer bone metastasis. The authors discussed many different markers of bone metastasis diagnosis as well. However, the text on exosomal microRNAs is very limited and does not mention many recent reports in this area, the authors might want to mention studies like Ruis-Plazas et al., 2021, Li et al, 2021, Patel G K et al., 2021, Fredsoe et al. 2019, Lorenc et al. 2020, etc in this section. Also, a table summarizing various published prostate cancer bone metastasis biomarkers with original reference for each study would be a good addition to the review article. Please double check for acronyms, some are not expanded at the first place at their first mention (ex: PCa, OC, OPN). Altogether, the manuscript is worth publishing with minor revisions.
Reviewer 2 Report
In the review article titled "Biomarkers for prostate cancer Bone Metastasis detection and prediction" by authors Mingshuai Ying et al., the authors summarized significance of biomarkers in prostate cancer accompanied with Bone metastasis. And summarized, many known biomarkers are widely used in clinical settings and others needed further clinical validation.
Overall the manuscript is well written and reads well however the manuscript lacks novelty to make it apart from the already published reviews which talk extensively about the similar points made by the authors.
Comments:
1) in MicroRNA biomarkers, miR466 is not discussed.
2) To improve the manuscript overall clinical value of each biomarker should be clearly stated
3) Separate section is needed to highlight how these biomarkers are relevant and exploited for developing therapy
4) To further validate the biomarkers as suggested by authors, a possible approach that can be used to do so will help improve the manuscript.
Reviewer 3 Report
The present review article, jpm-2292529 entitled: (Biomarkers for Prostate Cancer Bone Metastasis Detection and Prediction)
The current review article contained a good work, good analysis and written in a good way. However, I have some minor comments as following:
- In page (1) **** distal clinical symptoms. traditional tec….*** Traditional need to be Capital letter. In page (2) *****and osteoclast precursors are drawn to and active….*** drawn to is in complete , please check ???
- P value need to be italic along the whole review article.
- . [48].,. [51].,. [52].,. [55]. These are examples of improper form of references, please check along the whole manuscript – add dot only after reference numbers???
- There are some words with different fonts along the whole review article e.g: *** synthesized by osteoblasts [58]. Some**** , *** OPG in the bone microenvironment of their tibias. In contrast,..***
-In page (5)**** Additionally, there was a strong Interrelationship between serum TRACP-*** please check as it is one sentence ???
- In page (6) please check the format of : Wei et al[79] ,,,, and [91], And they found
